# The Impact of Coparenting on Mothers' COVID-19-Related Stressors

**Marsha Kline Pruett** [1,*], **Jonathan Alschech** [2] **and Michael Saini** [3]

1  School for Social Work, Smith College, Northampton, MA 01060, USA
2  School of Social Work, University of Northern British Columbia, Prince George, BC V2N 4Z9, Canada; jonathan.alschech@UNBC.ca
3  Factor-Inwentash Faculty of Social Work, University of Toronto, Toronto, ON M5S 1V4, Canada; michael.saini@utoronto.ca
*  Correspondence: mpruett@smith.edu

**Abstract:** To test and explore whether more positive coparenting will significantly predict lower COVID-19-related stress across family configurations and dynamics and across both higher- and lower-income mothers, we developed and circulated an online survey among mothers from the U.S. and Canada. Coparenting was measured using the Coparenting Across Family Structures (CoPAFS) short form (27 items) scale, comprised of factors representing five coparenting dimensions: communication, respect, trust, animosity, and valuing the other parent. Items specific to COVID-19 stressors assessed the types of stressors each parent faced. The sample consisted of 236 North American mothers, mostly white ($n$ = 187, 79.2%) and aged 30–50 years. The surveyed mothers reported a consistent and significant relation between more positive coparenting and less COVID-19-related stressors whether parents were living together or not, married or divorced, and with a lower or higher income level, suggesting the importance and centrality of positive coparenting as a key factor for family well-being. Coparenting was especially predictive among mothers who were never married and those with lower incomes.

**Keywords:** coparenting; COVID-19; mothers; stress; family structures





## 1. Introduction

The rapid onset of COVID-19 created an immediate necessity for families with children to acclimatize to dramatic changes in family life, including living, working, and learning at home, with minimal contact with their extended family, friends, and outside influences. Efforts to prevent or slow the spread of the virus have required families to adjust to new social distancing measures as part of the normal rhythms of everyday life. These safety measures, while critical to implement, have associated costs. Physical distancing measures contribute to increased feelings of isolation, stress, and anxiety, mood disorders, sleep issues, PTSD, and emotional burnout (Brooks et al. 2020; Lateef et al. 2021; Rajkumar 2020). Families demonstrate more intimacy and conflict, hence an increase in marriages, pregnancies, divorces, and intimate violence since the COVID-19 pandemic (Humphreys et al. 2020; Lateef et al. 2021; Moreira and Costa 2020), consistent in various types of disasters around the world (Cohan and Cole 2002; Gearhart et al. 2018; Kofman and Garfin 2020; Parkinson and Zara 2013; Peterman et al. 2020; Prasso 2020; Xu and Feng 2016).

In prior crises, research primarily focused on parents' mental health rather than the systemic importance of coparenting. The research centered on SARS, similar to COVID-19 in its airborne transmission, the speed with which it spread across countries, and the required infection control measures (Dwosh et al. 2003; Maunder et al. 2003; Patrick 2003). Robertson et al. (2004), for example, pointed to the relationship between infection control measures and the disruption of daily routines and parenting roles alongside competing career demands. Chan et al. (2007) conducted interviews with parents of 17 children during

the SARS outbreak. They found themes relevant to today's pandemic, including anxiety stemming from the challenges of juggling overlapping responsibilities between the child and home or work.

Concern during environmental crises is especially acute for parents with lower incomes or living unmarried or separated. Public health crises differentially affect public-housing residents, single-parent families, and low-income populations (Bouye et al. 2009; Mikolai et al. 2020). These groups were found to be more susceptible to complications from the general public health emergency due to insufficient funds to stockpile supplies, a lack of adequate health care and/or health insurance, unstable employment, and a lack of job benefits, weak social support networks, and the inability to effectively follow public health recommendations due to competing everyday survival needs.

### 1.1. COVID-19

Families with children have been found to be at greater risk of mental health problems during COVID-19 (Gunther-Bel et al. 2020), especially women and families with younger children and less education (Huebener et al. 2021). Lateef et al. (2021) completed a scoping review on the psychosocial consequences of COVID-19 on parents and children, indicating that parents with children tended to experience greater psychological maladjustment than adults without children. Children have not only been impacted by COVID-19 directly (e.g., less outdoor activities with peers), but they may have been negatively affected by the higher than usual levels of anxiety and stress exhibited by their parents, who also might be finding it challenging to cope with domestic issues such as parental conflict and disharmony.

Parental stress has increased children's risks in the pandemic in more subtle ways as well. Given the primary role of parents in supporting their children's coping and in creating a family environment that scaffolds coping responses (Luthar 2006; Fulton and Drolet 2018), parental stress may have limited the efficacy of parental support in response to the pandemic and related stressors. Parents have faced unique challenges, such as working cooperatively and adjusting to accommodate disruptions stemming from changes in school and employment schedules, the complexity of implementing social distancing among multi-household families, and negotiating differences in COVID-19 protocols between households (Lebow 2020). Family relationships have been intensely challenged and at times reshaped by the COVID-19 pandemic, with shared beliefs, communication, and closeness providing systemic buffers to external stressors (Prime et al. 2020).

During the pandemic, few studies have emerged with longer-term data. In one exception, Feinberg et al. (2021) conducted a longitudinal study on parents with children under 18 years old, examining differences in individual and family stress levels before (2017–2019) and later in the pandemic. Consistent with previous findings, a decline in both parents' and children's mental health was evident, specifically in parental depression ($d = 0.82$) and children's internalizing and externalizing behaviors ($d > 1$). There were small to moderate negative changes for both coparenting quality ($d = -0.40$) and parenting quality ($d = -0.24$), indicating a deterioration on multiple levels of family relationships after the pandemic. The implications drawn from the results include the potential benefit of coparenting in supporting families during stressful circumstances. Insufficient attention has so far been given to relations between coparenting and experiences of COVID-19 stressors across all family structures, whether intact or separated/divorced, or in families in which the parents were never together as a couple.

### 1.2. The Importance of Adding Coparenting to COVID-19 Research

Characterized by a sense of solidarity, a joint perspective and belief that "we are a team" with mutual engagement and shared labor distribution between caregivers (Pruett and Pruett 2009), coparenting is one of the hallmarks by which we can rely on promoting children and family well-being (Walsh 2016). It is not only a term for two-parent families; coparenting is defined as two or more adults in any family structure engaging in the shared

activities and responsibilities of raising a child (McHale and Lindahl 2011; Saini et al. 2019), and accounts for the environment in which children are raised. Quality coparenting sets the stage for promoting resilient family contexts—the very kind of environment that is needed in the face of a pandemic that disrupts every facet of daily life. Among the key processes that promote healthy family contexts is the parents' ability to form cooperative parenting/caregiving teams. These teams are most effective when they are able to sustain connectedness, mobilize social and economic resources, and keep communication channels open for sharing joy as well as pain (Walsh 2016).

Researchers have repeatedly found coparenting to be a predictor of parenting quality and family stability (Cabrera et al. 2012; Feinberg et al. 2016, 2021) across family structures (e.g., intact and separated), including sexually and gender diverse families with planned gay and lesbian coparenting strategies (Farr et al. 2019; Johnson et al. 2016). Positive coparenting optimizes parenting: positive adult relationships and coparenting are associated with better mother–child and father–child relationships in both middle-class and low-income families (Doss et al. 2014; Feinberg et al. 2016; also see Pruett et al. 2017). In intact couples, coparenting quality is associated with parenting quality even more closely than marital satisfaction (e.g., Feinberg et al. 2010). The positive effects of father involvement on child development are secured through positive coparenting (Pruett et al. 2017; Pruett and Pruett 2020). A differing focus on distinct coparenting qualities suggests presumed differences in coparenting across family structures, for example the emphasis on triangulation (Madden-Derdich et al. 1999), cooperation, and boundary ambiguity (Allen 2007; Beckmeyer et al. 2021; Pruett and Donsky 2011) in separated families over intact families. Coparenting support, too, drops off from high to low in separated families in a way that would not be expectable in intact families (Mallette et al. 2020). There have been almost no direct tests of coparenting across marital status (cf. Bronstein et al. 1993). However, a preliminary testing of this coparenting assessment tool indicates that the tool differentiates coparenting in couples living together versus apart (Saini et al. 2019). Given the importance of coparenting for partner outcomes, the presumed differences in coparenting across marital status, theoretically, and the very preliminary data showing that coparenting can be assessed differentially across marital status categories, there is a strong rationale for further research in this area.

Additionally, child outcomes are associated with positive coparenting across family structures and parental gender orientation (Farr et al. 2019; Neppl et al. 2019). Lifestyle disruptions from the current pandemic, like the disasters preceding it, will increase parental depression among some individuals, adults and children (Lai et al. 2013; Tang et al. 2014; Wilson-Genderson et al. 2018; Brown et al. 2020). Parental depression, both mothers' and fathers', is associated with increased depression in infants and children (Olfson et al. 2003; Merwin et al. 2017; Mueller et al. 2019), with fathers' sensitive parenting shown to moderate the effect of mothers' depression on children (Vakrat et al. 2017). Moreover, complex patterns of maternal and paternal depression and anxiety interactively affect parenting (Ierardi et al. 2019), with transactional patterns varying by family ethnicity and child gender rather than family structure (Tyrell et al. 2019). Supportive coparenting can directly serve as a buffer to the transmission of negative effects to children from depressed and anxious parents (Feinberg et al. 2016).

While coparenting has been virtually neglected in the professional literature on disasters, as we illustrated above, the buffering potential of positive coparenting became more obvious during COVID-19. When coparenting falters, so does the quality of parental care. For example, caregivers in India perceiving a high level of stress due to COVID-19 and having different discipline styles reported higher levels of parenting difficulties than usual, with greater likelihood of responding to children's misbehavior with punishments (Sahithya et al. 2020). COVID-19 presents a potential turning point for families, presenting a key opportunity to promote individual and coparental resilience in a vulnerable moment that stands at the vortex of crisis and opportunity.

### 1.3. Purpose of the Study

Based on general clinical wisdom and experience with other kinds of environmental disasters and health crises (Dwosh et al. 2003; Maunder et al. 2003), professionals and nonprofessionals alike have been providing various advice in both print and online media. This advice has varied in quality in terms of the attention paid to developmental differences among children, economic differences among families, and how the challenges parents face differ with the extent to which they are facing these challenges attuned and in league with each other or in continuing conflict. The specter of such immediate and encompassing changes has created an essential role for research to articulate the needs of families and to document the challenges and opportunities for growth as families strive to adapt to living in this pandemic environment and exercise healthy coping strategies. As such, some of the lessons learned from how families navigate and manage stressors during this pandemic may prove transferable to other societal crises.

### 1.4. The Study Model: Positive Coparenting as a Protective Buffer

Based on existing literature as well as practice wisdom and experience, we expected that coparenting as a latent construct underpinning five factors (Communication; Respect; Trust; Animosity and Valuing the other parent) would be negatively associated with COVID-19-related stressors. This association would hold across five specific concerns (coparent infects child; coparent infects mother; child will get COVID-19 unspecified; child going outside; child meeting other people). The study model is presented in Figure 1. Specifically, we hypothesized that:

1. More positive coparenting (a higher score on the CoPAFS scale) will significantly predict a lower COVID-19-related stress across all family configurations (unmarried, separated/divorced, married) and for both higher- and lower-income mothers.
2. The size of the effect of positive coparenting (indicated by the regression coefficient of coparenting as a predictor and COVID-19-related stressors as the outcome variable) and how much of the COVID-19-related stress it accounts for (indicated by the proportion of the variation in COVID-19-related stressors predicted by the model), though significant for all family configurations, will vary between these groups, and for both higher- and lower-income mothers.
3. The relative weight of each of the coparenting factors, though all significantly accounted for by coparenting as a latent construct, will vary across family configurations and dynamics and for both higher- and lower-income mothers.

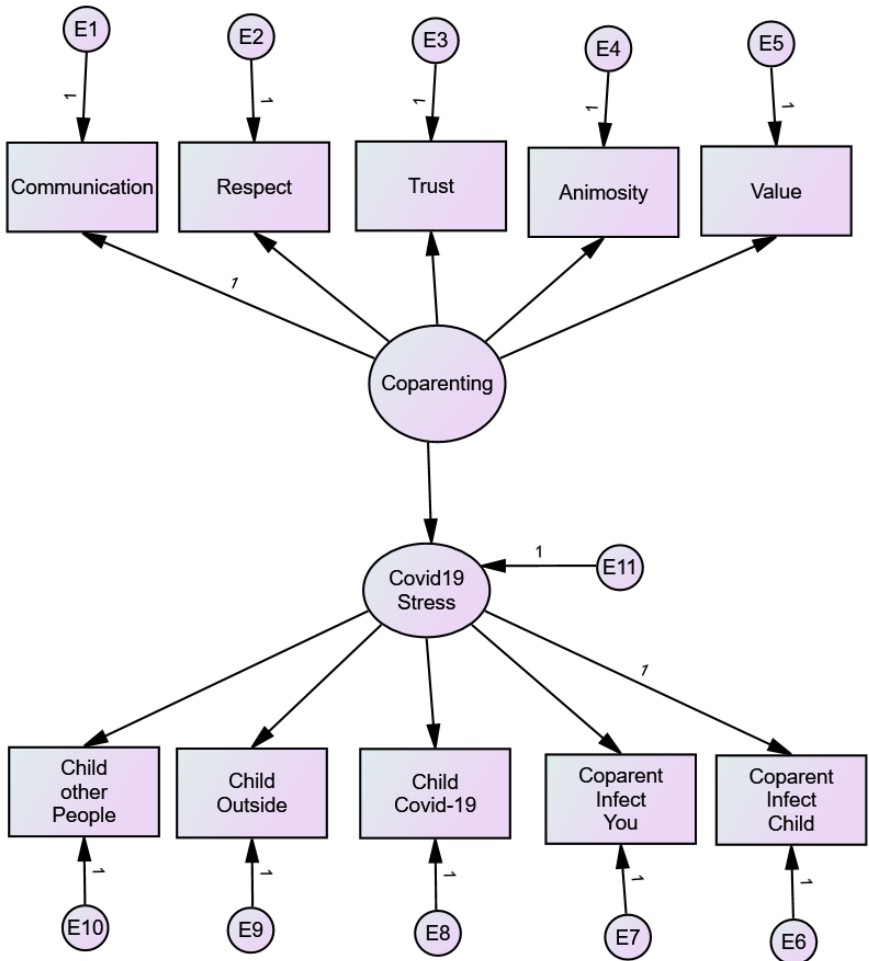

**Figure 1.** The study's model.

## 2. Materials and Methods

In order to test the study's model (Figure 1 above)—comparing the relations between coparenting and COVID-19-related stressors for mothers across family structures and income groups—we developed and circulated an online survey among mothers from the U.S. and Canada. In this study we analyzed the collected data so as to explore the commonalities and differences in coparenting in the context of the COVID-19 pandemic.

### 2.1. Sampling

We conducted an initial exploration of the theoretical model's fit across family structures and income levels, with the expectation that it would provide important insights and findings informing subsequent studies with more representative and diverse samples. We utilized convenience sampling through an invitation to answer an online survey concerning coparenting and COVID-19-related stressors, which was circulated in multiple online parenting groups and mailing lists between March 2020 and March 2021. The inclusion criteria for participation in the survey included: (1) self-identifying as a parent with a child under the age of 18 at the time of answering the survey; (2) self-identifying as sharing parenting with at least one other adult; and (3) being able to read English to complete the survey. Participants were not compensated for their participation in the survey. However, the primary investigators of this study offered free coparenting webinars for parents, circulating an invitation to answer the survey among those participating.

### 2.2. Data Collection

This study used an online survey as the method for data collection. The survey allowed for the distribution of the survey to potential participants in an online format, which was critical for collecting data during the pandemic. As the electronic survey was broadly circulated, participants were recruited from the United States and Canada. The electronic survey was created using the software "Survey Monkey". This software uses security technology such as firewalls and encryption to ensure the protection of data. The survey consisted of questions exploring the role, function, and effectiveness of parent coordination as well as pandemic-related challenges and stressors. The online survey took approximately thirty minutes to complete. We did not collect information about the participants' geographical location. Hence, no information about the local isolation regulations in force while answering the survey, or the local infection rates at the time of responding to the survey, were collected.

### 2.3. Measures

Coparenting was measured using the Coparenting Across Family Structures (CoPAFS) short form (27 items) scale. The CoPAFS-27 was created after a full review of the coparenting literature and tools, aimed at developing and validating a coparenting scale for which language and design does not assume gender-specific parental roles or the family structure (Saini et al. 2019). The scale is comprised of 5 factors capturing coparenting dimensions: communication, respect, trust, animosity, and value of coparenting. Each statement was measured by a 5-point Likert-scale ranging from 1 = Strongly Disagree via 3 = Neutral to 5 = Strongly Agree. Examples of items include: "It is important that my child loves both parents"; "I value the other parent's parenting skills"; "I work well with the other parent when decisions need to be made about our child"; and "I find it difficult to support the other parent's relationship with our child". Items specific to COVID-19 stressors assessed the types of stressors the parent faced; daily life disturbances; coparenting in the past two weeks; worries such as (1) the extent to which the mother felt worried that the child will contract COVID-19; (2) the extent to which the mother felt worried that the coparent will infect the child with COVID-19; (3) the extent to which the mother felt worried that the coparent will infect her with COVID-19; (4) the extent to which the mother felt worried about the child going outside during the pandemic; and (5) the extent to which the mother felt worried about the child meeting other people in person during the pandemic; and a general evaluation of how the pandemic was affecting parenting and coparenting.

Both the COPAFS SF scale and the COVID-19 items demonstrated good or excellent internal reliability, as shown in Table 1.

**Table 1.** Measures of central tendency and internal validity indices.

|  |  | Mean (SD) | Median | Cronbach's Alpha |
|---|---|---|---|---|
|  | Total | 80.36 (21.94) | 76.00 | 0.93 |
|  | Communication | 12.89 (5.17) | 12.00 | 0.82 |
|  | Respect | 10.58 (4.40) | 10.00 | 0.81 |
| CoPAFS | Trust | 20.30 (6.30) | 19.00 | 0.80 |
|  | Animosity | 18.45 (5.11) | 18.00 | 0.75 |
|  | Value | 18.10 (4.61) | 18.00 | 0.77 |
| COVID-19 related stressors |  | 16.37 (4.89) | 17.00 | 0.82 |

### 2.4. Data Analysis

The data were exported from "Survey Monkey" to the statistical software SPSS (version 26), including its structural equation modeling extension package AMOS (version 26), for analysis. Descriptive and bi-variate analyses were performed, and the Cronbach's alpha coefficients were calculated as a measure of the internal validity of the scales used. We used

structural equation modeling with a maximal likelihood estimation method, assessing how well the study model fitted the data. Comparing model fits, regression coefficients and the proportion of variation accounted for mothers reporting (1) living together with the other parent; (2) separated/divorced; or (3) never together with the coparent. Model fit indices, regression coefficients, and the proportion of variation accounted for were also compared between higher income and lower income participants.

The aim of assessing how well the model fitted the data, specifically comparing model fit indices and regression coefficients across the different 6 groups of interest in the sample, was exploratory. Comparing and contrasting the differences between the 6 groups, generated various insights, as discussed below, informing the design of future steps in the research program. Hence, rather than seeking to establish the robust invariance of our model across different samples (as in multigroup confirmatory factor analysis), our analysis sought to describe and explain the data collected, with the intention of generating hypotheses for subsequent confirmatory studies.

Following Kline (2016), the model fit indices calculated and reported were, firstly, a chi-squared test indicating the difference between observed and expected covariance metrics. The P value of the chi-squared test should be above 0.05 (not significant). Although this is strongly influenced by the sample size, this may be misleading in either small samples (leading to the acceptance of an inappropriate model) or large samples (leading to the rejection of appropriate models). The second type of model fit indices calculated and reported was the root mean square error of approximation (RMSEA), which measures the discrepancy between the hypothesized model, with optimally chosen parameter estimates, and the population covariance matrix. Thirdly, the root mean square residual (RMR) shows the square root of the discrepancy between the sample covariance matrix and the model covariance matrix. The Goodness of fit index (GFI) is a measure of fit between the hypothesized model and the observed covariance matrix. The normed fit index (NFI) analyzes the discrepancy between the chi-squared value of the hypothesized model and the chi-squared value of a null of baseline model in which all the variables are assumed to be uncorrelated. Comparative fit index (CFI) analyzes the model fit by examining the discrepancy between the data and the hypothesized model, while adjusting for the issues of sample size inherent in the chi-squared test of model fit.

## 3. Results

### 3.1. Demographics

The final sample consisted of 236 North American mothers, mostly 30–50 years old: 110 (46.6%) were 30–39 years old; 87 (36.9%) were 40–49 years old; 21 (8.9%) were 20–29; and 18 (7.6%) were older than 50. Mothers of color were highly underrepresented in the sample ($n = 12$; 5.1%); the largest group of non-white mothers were Latina ($n = 16$; 6.8%). Most participants were white ($n = 187$, 79.2%). The majority of the mothers in the sample, 137 (58.1%) reported working full time during the past 6 months, 39 (16.5%) reported being employed part time; 27 (11.4%) reported they were looking for a job, and 7 (3%) were on social assistance. The annual pre-tax household income reported by the mothers in the sample is summarized in Table 2, showing that the sample was relatively diverse in terms of socio-economic status.

**Table 2.** Study sample's reported annual income.

|  | Frequency | Percentage |
|---|---|---|
| Less than 30K | 42 | 17.9% |
| 30–59K | 65 | 27.7% |
| 60–89K | 44 | 18.7% |
| 90–119K | 31 | 13.2% |
| 120K+ | 53 | 22.6% |

The sample included mothers living in different family structures: cohabiting with the coparent, separated/divorced, and never lived with the coparent. Yet mothers of intact families were underrepresented (*n* = 42; 17.8%), with the majority (*n* = 132; 55.9%) being divorced/separated and 62 (26.3%) never having lived with the coparent of their child.

### 3.2. Model Fit Analysis

The study model's fit to the data (full sample) and to 5 subgroups in the sample—(1) mothers of intact families; (2) divorced/separated mothers; (3) mothers who never lived with the coparent for their child; (4) mothers with a household pre-tax annual income of above 60K; (5) mothers with a household pre-tac annual income below 60K—were assessed using structural equation modeling with maximal likelihood estimations. The model fit indices for the 6 samples are summarized in Tables 3 and 4; the regression coefficients and the proportion of the variation predicted for all variables in the model into each of the 6 samples are presented in SI Figures S1–S6.

**Table 3.** CFA model fit indices for the 6 samples.

|  | Chi-Square | DF | P | RMR | GFI | CFI | NFI | RMSEA |
|---|---|---|---|---|---|---|---|---|
| Full sample | 239.70 | 34 | <0.01 | 0.53 | 0.83 | 0.85 | 0.83 | 0.16 |
| Divorced/Separated | 181.34 | 34 | <0.01 | 0.61 | 0.78 | 0.79 | 0.76 | 0.18 |
| Never married | 67.18 | 34 | <0.01 | 0.70 | 0.82 | 0.82 | 0.70 | 0.12 |
| Intact | 74.09 | 34 | <0.01 | 0.49 | 0.79 | 0.86 | 0.78 | 0.17 |
| Income > 60K | 163.57 | 34 | <0.01 | 0.54 | 0.80 | 0.85 | 0.82 | 0.17 |
| Income < 60K | 134.21 | 34 | <0.01 | 0.74 | 0.78 | 0.80 | 0.75 | 0.16 |

**Table 4.** Predicting COVID-19-related stress for the 6 samples.

|  | Full Sample | Divorced/Separated | Never Married | Intact | Income > 60K | Income < 60K |
|---|---|---|---|---|---|---|
| Proportion of variation on COVID-19 Stress | 0.07 (7%) | 0.07 (7%) | 0.16 (16%) | 0.05 (5%) | 0.06 (6%) | 0.13 (13%) |
| Regression coefficient Coparenting → COVID-19 Stressors | −0.27 ($p < 0.01$) | −0.26 ($p < 0.01$) | −0.40 ($p < 0.01$) | −0.22 ($p < 0.01$) | −0.25 ($p < 0.01$) | −0.35 ($p < 0.01$) |

The model fit indices indicate a similar overall model fit in each of the 6 samples. Though the model fit indices were less than ideal, all regression coefficients within the model were significant in all 6 samples. This lends an encouraging initial support to our first hypothesis, positing that a higher score on the CoPAFS scale significantly predicts lower scores on the COVID-19-related stressors scale, across all family configurations and for both higher- and lower-income mothers. While fully confirming the model will require testing on more representative samples, the similarity in the overall model fit indices for all 6 subsamples allowed us, as detailed below, to concentrate on the differences in results when testing the model across the different family structures and income levels.

As posited in our second hypothesis, the regression coefficient of coparenting as a predictor of COVID-19-related stressors, though significant for all samples, varied considerably, ranging between −0.22 for mothers in intact families and −0.40 for mothers who never lived with the coparent of their child. In other words, the effect of coparenting on COVID-19-related stress was almost double for mothers who never lived with their child's coparent in comparison to mothers who are still together with the other parent of their child. The proportion of the variance in COVID-19-related stressors accounted for by coparenting, though significant for all family structures and for mothers with both higher and lower incomes, varied considerably, ranging between merely 5% of the variation for

mothers in intact families to 16%—more than three times greater—for mothers who never lived with their coparent.

As posited in our third hypothesis, the regression coefficients for each of the five factors composing the CoPAFS scale were significant for all six samples, yet the coefficients and the proportion of the variation in each of the factors accounted for by the underlying latent construct (coparenting) varied considerably across family configurations and between mothers with higher versus lower incomes. These variations indicated the differences in the relative contributions made by each of the factors to the overall quality of the coparenting relationship for each of the samples. While communication was the most important factor for mothers with an annual income lower than 60K, trust and respect were the most important factors for mothers with an annual income higher than 60k. Communication and respect were the most important factors for mothers in intact families as well as for divorced/separated mothers, while trust was the most important factor for mothers who never lived with the coparent of their child.

## 4. Discussion

This is the first known study to begin exploring the impact of coparenting on mothers in the U.S. and Canada during the COVID-19 pandemic, across family structures and between higher- and lower-income levels. Especially noteworthy is the comparison we present between intact, separated/divorced, and never-together families, centering coparenting as a key factor in family coping, exploring the different relative importance of the factors composing the construct of coparenting between different family structures and income levels.

The surveyed mothers reported a consistent and significant relation between more positive coparenting and less COVID-19-related stressors. To be sure, this is merely an association identified in the context of a cross-sectional study on a convenience sample, and hence it is in no way sufficient evidence for determining a causal relationship. However, our findings suggest it is highly plausible that positive coparenting served as a protective buffer against COVID-19-related stressors for mothers, regardless of their family structure and income stratification. This may be further extrapolated to suggest that positive coparenting may be a protective buffer for other comparable general-society (as opposed to personal family-specific) stressful times and situations. However, evidence of this effect is unknown to date. The protective power of positive coparenting has been demonstrated by previous research for intact families (Feinberg et al. 2010; McHale et al. 2019; Pruett et al. 2017), divorced families (Pruett et al. 2011), as well as families with low incomes or who are living in poverty (Feinberg et al. 2021; Jamison et al. 2017). In this study, we have provided some initial encouraging evidence about the benefits of positive coparenting for maternal wellbeing in the context of the current COVID-19 pandemic.

The findings from this study lend preliminary and encouraging evidence for the importance and centrality of positive coparenting as a potentially key factor in maternal wellbeing in the face of COVID-19, which may in turn increase and support family resilience (McRae et al., in press). This finding may also be interpreted as supporting current research showing that poor coparenting relationships are related to higher maternal stress, not only in high conflict-separated/divorced and/or low-income families, but also for those living in intact families or those reporting higher incomes (Kang et al. 2020). Positive coparenting is a potent protective factor, which can be successfully practiced across all family structures. Separated/divorced families, as well as families in which the coparents have never been romantically involved, may nevertheless practice and benefit from positive coparenting, thriving during times of increased stress and overall societal hardships.

Our study found that mothers who have never been romantically involved with the coparent of their child benefitted most from positive coparenting—almost twice as much as mothers in intact families. This was also shown by our finding that coparenting accounted for three times more of the variation in COVID-19-related stressors as reported by never-together mothers in comparison to those reported by mothers in intact families. These

findings reflect the increased stress and challenge for families headed by a single mother during the pandemic, and the great difference brought about by a positively involved coparent. While the stress experienced by families headed by a single parent and more so by a single mother have received attention, especially in other countries (Craig and Churchill 2021; Febrianto 2021), in the U.S. and Canada, the significant, even dramatic, difference in stress made by positive coparenting in these contexts is an understudied aspect.

Similar to our data, Russell et al. (2020) also found that parents with unmet financial needs would experience a higher caregiver burden. They also reported that men rather than women experience a greater burden, and single parents experienced lower depression and anxiety than separated/divorced parents. Both of the latter findings point to the potential of strengthening coparenting as a buffer for the burden experienced by families. Family structure clearly plays a role in familial coping that is important to learn more about in subsequent research. Fathers may experience more caregiver burden if they are the ones to manage more employment stress at home, compared to mothers who might have done more of that juggling pre-pandemic. Furthermore, the change in communication, child rearing responsibilities, and schedules might have been greater for separated/divorced parents, or require more negotiation to shift. In either case, how the partners work out the change will likely have implications for the entire family's well-being.

Moreover, parents who are depressed, anxious, and overwhelmed—especially those with economic hardship—are likely to have more difficulties parenting (Goodman et al. 2017; Newland et al. 2013), and coparenting can be a source of support and buffering when one parent has neared the end of their patience and ability to focus on their child. We also need to understand what having two overwhelmed parents means under conditions of different economic strain, as well as high- and low-quality coparenting, which could be expected to have differential effects among coparents living together rather than apart. Interestingly, parents living apart have participated more readily in some studies of co-parenting stress than those living together (e.g., Saini et al. 2019), so perhaps it is a more pressing issue for them.

Our results suggest, not surprisingly, that higher-income families tend to be less impacted by the pandemic than lower-income families. Furthermore, mothers from lower-income families might have either a heavy workload or the need to provide a larger quantity of maternal care due to their inability to afford daycare or babysitting. This would be true across family structures, although the role of coparenting in alleviating some of the strain awaits further investigation.

While positive coparenting was a significant protective buffer for all family structures, the relative importance of the factors composing the construct of coparenting varied between family structures and income levels. For example, communication was a more important factor for couples than for those parents who never lived together, possibly indicating families whose discord increased during the pandemic. However, trust was the most important factor in parents who never lived together, suggesting that without prior romantic involvement, the building and securing of trust between coparents may be more elemental than communication. Fostering trust, respect, and communication among co-parents, including recognizing and directly addressing families' potential relational patterns that emerge as a result of either increased or decreased time spent together, may also improve the mental health of caregivers (Lebow 2020).

Similarly, support with establishing boundaries between family members, particularly among members who may be spending more time than usual in the home space due to the cancellation of out-of-home programming and/or working from home, may be beneficial to parents' improved mental health (Lebow 2020). Communication was the most important factor for intact, separated/divorced, and lower-income families, perhaps implying that the trust was present, but communication is needed to overcome economic and relationship challenges created by the pandemic.

For parents living apart, parenting plans and agreements are designed to effect consistency and predictability in family members' lives, but they cannot include emergency measures for situations like the COVID-19 pandemic. This leaves gaps in guidance and care from the courts when families are faced with crises that require a quick adaptation from existing legal agreements that are no longer as relevant to the structures set up historically to maintain their co-parenting relationship. In many cases, this resulted in restrictions of contact (often between fathers and children) to reduce the transmission risk among families who had worked hard in the past to develop a balanced parenting schedule. Reduced court availability exacerbates these disruptions, as families found it challenging to amend their court agreements, with or without the legal structure. Given the ongoing nature of the pandemic, temporary court structures that are nimble enough to pivot and respond quickly and flexibly are necessary. As more courts have tackled meetings online, they are finding ways to be responsive while maintaining a closed courtroom.

## 5. Limitations

One significant limitation of this study is that participants comprised a convenience sample rather than a representative one, greatly limiting the generalizability of the findings. To date, the sample has not been sufficiently diverse. The respondents were largely mothers from separated families. Moreover, the respondents were mostly white, heterosexual, and educated, with a large proportion of employed mothers, such that the results reflect a privileged lens and may fail to represent the experiences of BIPOC and sexually diverse (LGBTQ+) families. Further, this analysis may not accurately portray the realities of low-income and/or rural co-parents, since most respondents were from urban settings, and the relatively high educations levels may suggest that the low-income group in our sample was not typical. A further limitation is that the survey asked only basic questions about the parents' relationship before the pandemic started, and so there is an absence of a strong baseline with which to compare dynamics during the pandemic. Different states in the USA and provinces and territories in Canada varied in their infection rates and physical isolation measures, and as we could not collect information about the precise location of the respondents (beyond living in the US or Canada), we were unable to control for the effects of these variations in COVID-19 measures.

## 6. Future Research

This research, though preliminary, suggests that focusing solely on mothers will negate the greater burden fathers may be feeling, perhaps with less support from their employment and society more generally, for adapting work hours and schedules to family needs during the pandemic. Despite the increased awareness of the contributions of fathers in the lives of children, fathers still receive less support in their parenting than do mothers across societal institutions (Pruett and Pruett 2020). Not only should we be paying more attention to the needs of fathers as parents, but also as coparents, since their positive involvement improves the lives of mothers and children (Pruett et al. 2017). To study coparents with full integrity, paternal responses must be sought in numbers that approach those provided by mothers in order to better examine couple differences as well as gender distinctions.

Future research should include a more extensive questioning about co-parenting's history and a broader set of communities and identities. We need to know what types of resources would help different co-parent subgroups recover, and specifically what kind of online mediation resources could be made available, at little to no cost, on a wider basis, for co-parents experiencing increased conflict due to the pandemic. Employing additional strategies in future questionnaire collection in order to attract more diverse participant populations (e.g., sexual orientation, race, ethnicity, social economic status) is needed.

The pandemic shows little signs of abating, highlighting the continued urgency in stemming some of the negative impact on families and increasing wellbeing. Enhancing coparenting information, support networks, and services and interventions offer promising

avenues for helping families emerge stronger and in better unity when the pandemic's virulence finally ebbs.

**Supplementary Materials:** The following are available online at https://www.mdpi.com/article/10.3390/socsci10080311/s1, Figure S1: full sample. Figure S2: intact families. Figure S3: never together. Figure S4: Divorced /separated. Figure S5: income above 60K. Figure S6: income below 60K.

**Author Contributions:** Conceptualization, M.K.P., J.A. and M.S.; methodology, Alschech and Saini; formal analysis, Alschech.; investigation, Pruett, Alschech and Saini; resources, Pruett; writing—original draft preparation, Pruett; writing—review and editing, Pruett, Alschech, Saini. All authors have read and agreed to the published version of the manuscript.

**Funding:** This research received no external funding.

**Institutional Review Board Statement:** The study was conducted according to the guidelines of the Declaration of Helsinki, and approved by the Institutional Review Board of Smith College (protocol code 19-036 and 23 March 2020.

**Informed Consent Statement:** Informed consent was obtained from all subjects involved in the study by their clicking on the survey page indicating their consent. No additional consent was collected because the online survey was anonymous.

**Data Availability Statement:** Data reported in this study can be obtained by contacting the corresponding author.

**Conflicts of Interest:** The authors declare no conflict of interest.

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
