# Peer review of "The Impact of Coparenting on Mothers’ COVID-19-Related Stressors"

_socsci, doi:10.3390/socsci10080311_

Round 1

Reviewer 1 Report

This paper has strong relevance to current world events and the field of family studies.  However, its results are weak and lend skepticism as to its empirical value.  Here are some suggestions for improvement:

1) In Section 1.1, the third paragraph addressing unmarried, separated, or low-income parents does not address prior pandemics and thus does not belong in that section.  The fourth paragraph addresses the current pandemic and thus belongs in the prior section.  The literature review might flow better if the information on prior pandemics was presented first to frame later information about COVID.

2)  In lines 122-125, it is asserted that there are positive effects of coparenting because the mother's relationship with the father is of positive quality.  The quality of the mother's relationship with the father is not the same concept as whether they cooperatively work together to parent their child.

3)  In the first paragraph in Section 1.4, an example is given in which contradictory information is stated that fathers report decreases in family satisfactions and then, "their family satisfaction increased from the change in COVID-19-related work circumstances."  It is unclear whether there was an increase or decrease in family satisfaction.  The example given as illustrating adaptive capacity is unclear, as the results cited do not have to do with adaptive capacity.

4)  As the article addresses differences between unmarried, separated/divorced, and married families in coparenting as one of its hypotheses, differences in coparenting between these types of families should be addressed in the literature review.

5)  In Section 2.3, example items for the COVID-19 stressors should be given.

6)  In Section 2.4, the paper does not simply do Confirmatory Factor Analysis, as that would involve validating a measurement model.  The authors use several structural models to determine relationships between coparenting and COVID-related stressors, so the methodological approach should be labeled structural equation modeling.

7)  A better approach to data analysis (and the one that is traditionally used when comparing across demographic subgroups) is to run one model with all subjects and then a multigroup CFA with all paths released and conduct a chi-square difference test to see if there is a difference in model fit for different model subgroups.  Paths can then be released to see where the differences are depending on if there is a significant difference. 

8)  None of the models have adequate fit.  All GFIs, CFIs, and NFIs are below .90, which is a liberal cutoff for those values that the model statistics should be over.  The RMSEA needs to be at least .08 or lower, a criterion which none of the models meet.  These facts are not mentioned anywhere in the paper, either in the results or discussion section, or how they limit the validity of the results of the model.  If this paper is to be accepted, a major discussion of why we should still find this paper scientifically valid and of merit despite these fit statistics should be included in the discussion or limitations section.

9)  The paragraph from lines 400 to 411 presents content that disagrees with the immediately previous content regarding the difficulties experienced by single mothers.

10) In several places, single mothers are references as never having resided with or had prior romantic involvement with (line 434) the coparent.  However, it is likely that the single mothers did have prior romantic involvement with and may have lived with the coparent in the past but do not currently.  Therefore, some of the interpretations regarding single mothers in the discussion may be inaccurate.

11) In the Limitations section, the generalizability limitations of having few married families and the majority of mothers being employed were not mentioned.

Author Response

We would like to thank the Reviewer for the critical and helpful comments made on the manuscript. W have made a number of changes based on these comments, addressed point-by-point in the attachment.

Reviewer 2 Report

Thank you for the opportunity to review this paper. Co-parenting is a topic that is understudied in the field and warrants attention. Overall the concept of the paper was  good, but I feel that the measures used and the conclusions drawn do not match. I am concerned that your measure of COVID stress does not adequately relate to co-parenting factors. I could be wrong, but I don't believe that quality of co-parenting would influence whether a mother worries about a child contracting covid-19 or her partner will infect her with covid-19. I suspect other factors such as rates of infection in the community, whether or not children go to school, where the partner works would play a bigger role in this. The paper is also unnecessarily long, and could be improved with major editing. A clear aim is needed. The content needs to focus on the aim of the study and not consider all research related to co-parenting. I have included more specific comments for consideration below. 

Abstract

Line 5- shouldn’t it be ‘in’ rather than and both higher and lower income mothers…

Intro

More context needs to be given about your study setting, were participants in lockdown at the time, what were the rates of covid infection, were schools closed etc..

Line54-57, needs rewording for clarity

Line 81- Not sure why environmental crisis is referred to here?

Line 145-46- Don’t think ‘brought in to sharp relief’ can be stated here

Line 151-161- Lacks clarity and could be removed

Line 173- satisfactions should be satisfaction

Line 195-197- avoid the word scrambling. Give advice about what in print and social media?

195-205- it is not clear what the overall aim of the study is…this section could be shortened considerably to 1-2 sentences

Aims and hypotheses do not consider income which is a focus of your study.

Measures

Line 252-254, did you develop this scale? Why did you not use a validated tool?

It is not clear how co-parenting factors would relate to concerns regarding a child contracting covid, the partner infecting family members, the child going out etc…

Table 1 if you used a 5 point scale why are the medians over 5?

Data analysis

Line 286, should be 0.05?

Line 286-299, references to support these approaches are needed.

Demographics

The majority of this text could go in a table.

Table 4 needs p-values

Discussion

Line 377-378- you cannot state that the findings suggest that coparenting as a key factor for resilience, when you did not measure resilience.

Line 379- why would the relationship be more acute for fathers?

Line 390- why is it that the relationship wasn’t as strong in intact families?

Line 403- partners parents?

Line 401- you did not measure caregiver burden

Line 408-409- partners parents?

Line 447-449- you did not measure depression and anxiety in the sample

Limitations

What about the lack of fathers in your sample?

Your questionnaire was quite limited and did not consider other influences on COVID-19 stress such as number of cases in the local area, or family members affected, whether the family was in lockdown or children could still attend childcare/school…etc

Author Response

(The authors gave the same response as above.)

Round 2

Reviewer 1 Report

The changes made to the paper were adequate in responding to some of the reviewer's concerns.  However, some concerns still need to be addressed if the paper is to be accepted:

1)  The authors stated that they added literature addressing coparenting differences between unmarried, separated/divorced, and married families.  This reviewer did not see any literature addressing how these different types of families performed coparenting activities similarly and/or differently.

2)  In their response to this reviewer, the authors state that their revised methods section clarifies their methodological strategy.  The data analysis section does not contain any significant revisions in terms of explanations.  The authors now correctly call their analysis "structural equation modeling" rather than CFA but do not explain why simply listing the different models in a table is a better approach than comparing them through a multi-group CFA.  There is also no clarification of how the "interpretive strategy" justifies the poor fit indices, although the fact that the fit indices are poor is now briefly acknowledged.

3)  In the Limitations section, generalizability is added, but not the specific issues of having few married families or the majority of mothers being employed.  Those specifics need to be addressed.

Author Response

We have received requests for minor changes in our manuscript socsci-1269038. We have made the requested revisions, as noted below.

  • Reviewer 1 asked for additional literature addressing coparenting differences across unmarried, separated and married families. Although we had addressed that in the previous revision, we added content which can be found on pg. 4, lines 151-163. Seven new references were added to the paper and bibliography.
  • Reviewer 1 also asked for further clarification regarding the methodological strategy. Those changes have been added and can be found on pg. 9, lines 357-364.
  • Finally, an addition was requested by the reviewer in the limitations section, noting that the respondents comprised a sample of mostly women (especially employed) and separated partners. This was added on pg. 15, lines 567-569.

In addition, we made some minor changes for clarity, especially in the abstract section.

Thank you for the opportunity to refine our paper.

Reviewer 2 Report

I feel your paper will make a good contribution to the field. You have clarified aspects that needed to be addressed, and the overall clarity of your paper is improved.

Author Response

(The authors gave the same response as above.)
